# SINQ: S̲inkhorn-N̲ormalized Q̲uantization for Calibration-Free Low-Precision LLM Weights

## Abstract

Post-training quantization has emerged as the most widely used strategy for deploying large language models at low precision. Still, current methods show perplexity degradation at bit-widths $\leq 4$, partly because representing outliers causes precision issues in parameters that share the same scales as these outliers. This problem is especially pronounced for calibration-free, uniform quantization methods. We introduce SINQ to augment existing post-training quantizers with an additional second-axis scale factor and a fast Sinkhorn–Knopp–style algorithm that finds scales to normalize per-row and per-column variances, thereby minimizing a novel per-matrix proxy target for quantization: the matrix imbalance. Our method has no interactions between layers and can be trivially applied to new architectures to quantize any linear layers. We evaluate our method on the Qwen3 model family and DeepSeek-V2.5. SINQ improves WikiText2 and C4 perplexity significantly against uncalibrated uniform quantization baselines, incurs a $0 - 2\%$ compute overhead, and can be further enhanced by combining it with calibration and non-uniform quantization levels. Code is available in the supplementary.

## 1 Introduction

Post-training quantization (PTQ) is a powerful approach to reducing the cost of neural network inference. Weight quantization reduces the storage, memory, and data movement required to run a neural network. As such, it is useful on its own whenever any of these components bottleneck the performance of an inference system. When integer (INT) or floating-point (FP) weight quantization is further combined with INT or FP activation quantization, it can also be used to reduce compute requirements by executing MatMul operations at low-precision. Potential speed-ups are substantial: For example, moving from bfloat16 to int4 weights yields a potential speedup of 4x in memory-bound scenarios. Weight-only quantization is especially popular in LLM deployment because accelerator memory capacity and data movement are often the initial performance bottlenecks in this scenario.

In this paper, we demonstrate that a carefully chosen uncalibrated, uniform quantizer can approach the end-to-end output quality of calibrated quantizers or non-uniform formats while being appreciably simpler: Calibration (and even more so end-to-end optimization) is an intuitive approach to improving the output quality of quantized models, but comes with the inherent downsides of possible bias and overfitting (Lin et al. (2024b)) and additional compute time required at quantization time (for models under large-scale deployment, this is not concerning as the quantization cost can be amortized over time, but for small-scale scenarios, this cost can be prohibitive). Similarly, non-uniform formats can offer an improvement over integer quantization (Dettmers et al. (2023)), but require potentially costly look-ups during inference and cannot be combined with activation quantization in compute-limited scenarios. In brief, if uncalibrated uniform quantization were to reach the same output quality, it would be preferable for these reasons. This paper takes a step towards closing the gap between these different approaches to quantization.

The key contributions of this paper are:

- We propose adding a scaling factor along the second axis of to-be-quantized matrix tiles.

- We propose a new proxy metric for ease of quantization of a matrix, the matrix imbalance (Eq. 4).

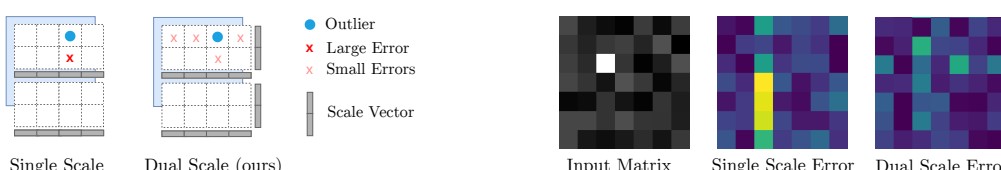

Figure 1: If we have scales along both dimensions of a matrix that is to be quantized, we can trade off the impact of outliers between rows and columns, which is impossible in single-scale quantization. Left: Conceptual illustration of error distributions with single or dual-scaling. Right: Example on small matrix.

- We propose a fast algorithm based on Sinkhorn-Knopp iterations for finding these dual weight scales to minimize the matrix imbalance (Sec. 2.2.1).
- In numerous experiments across different model scales, we show that our method improves over state-of-the-art baselines for calibration-free quantization methods.
- We provide code for easy quantization of LLMs using linear layers.

## 2 METHODS

We divide our method into two parts: Firstly, the quantized parameterization, i.e. the mathematical expression used to map between the full precision and the quantized matrix. All quantization methods used in practice, have some set of auxiliary parameters to use in this mapping. Secondly, the representation space, i.e. the space in which we instantiate the full precision matrix when quantizing it.

### 2.1 QUANTIZED PARAMETRIZATION

Typically, one does not simply replace the weight matrix with, for example, an INT4 matrix, but rather divides it into tiles and assigns some higher-precision auxiliary parameters to each tile. Here, we describe different possibilities for the type of auxiliary parameters to use and how to tile the matrix.

#### 2.1.1 PARAMETERIZATION PER TILE

**Scales + Shifts**  The most widely used approach uses a scale and a shift vector (e.g., Badri & Shaji (2023)), like so:

$$\mathbf{W}_{\text{approx}} = \vec{s} \odot (\mathbf{Q} + \vec{z}) \tag{1}$$

where $\mathbf{W}_{\text{approx}}$ is a $N \times M$ matrix (or matrix tile), $\vec{s}$ is a $N \times 1$ vector, $\vec{z}$ is a $N \times 1$ vector and $\mathbf{Q}$ is a quantized $N \times M$ matrix. Also, the transpose of this with $1 \times M$ vectors is commonly used.

**Dual-Scales**  In this paper, we propose a new parameterization based on an idea we call dual-scaling: Given a matrix (or a tile of a matrix), instead of supplying a single vector of scales along one dimension of the matrix, we supply two vectors, one along each dimension. Formulaically, we propose:

$$\mathbf{W}_{\text{approx}} = \vec{s} \odot \mathbf{Q} \odot \vec{t} \tag{2}$$

where $\vec{s}$ is a $N \times 1$ vector, $\vec{t}$ is a $1 \times M$ vector and the rest is as above.

The key benefit of Eq. 2 can be illustrated as follows: Say $W_{ij}$ is an outlying large value. By scaling up $s_i$ and scaling down $t_j$ we can trade off quantization errors that will occur in row $i$ for errors in column $j$. See Fig. 1 for an illustration.

**Dual-Scales + Shifts**  If we do not mind the potential additional overhead (or rather, if an accuracy improvement justifies it), we can also add shifts to the dual scales:

$$\mathbf{W}_{\text{approx}} = \vec{s} \odot (\mathbf{Q} + \vec{z}) \odot \vec{t} \tag{3}$$

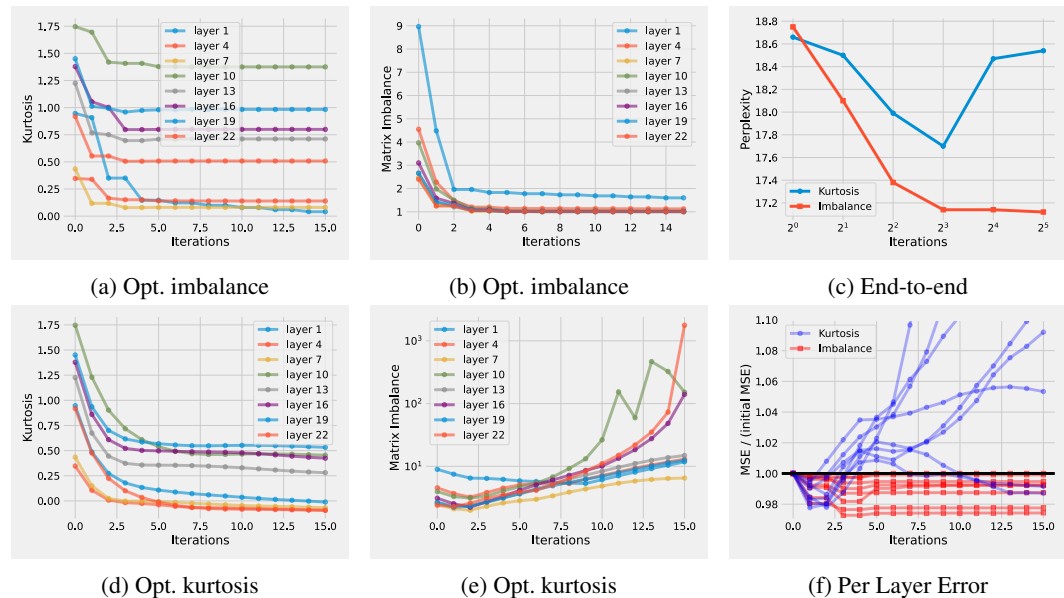

(a) Opt. imbalance      (b) Opt. imbalance      (c) End-to-end

(d) Opt. kurtosis      (e) Opt. kurtosis      (f) Per Layer Error

Figure 2: Results on Qwen3-1.7B. Minimizing the imbalance with our algorithm (a and b) decreases both the imbalance and the kurtosis. Minimizing the kurtosis directly with gradient descent (d and e) yields lower kurtosis, but causes a large imbalance; note the log-scale on (d). Finally (c) and (f) show the end-to-end perplexity on wikitext2 and per-layer RTN MSE improvement when optimizing imbalance or kurtosis, respectively.

### 2.1.2 TILING

Typically, (e.g., Badri & Shaji (2023); Lin et al. (2024b)) tiling for quantization is implemented along one dimension of the matrix that is to be quantized. By consequence, these tiles have *rectangular* shapes; e.g., a $N \times M$ matrix tiled with tile-size $T$ would yield tiles of shape $N \times T$. This could cause a problem with the dual-scale parameterization. Namely, the standard parameterization has $2 \times N \times M/T$ scale and shift parameters, while the dual-scaled only has $N \times M/T + M$.

To ensure that the dual-scale parameterization has approximately the same number of additional parameters, we can use a *2D* tiling that divides the $N \times M$ matrix into *square* tiles, e.g., of shape $T \times T$. For square matrices, this yields the same number of auxiliary parameters as the single-scale + shift approach with rectangular tiling.

Alternatively, we may use dual-scale parameterization together with a shift (as in Eq. 3). With 1D rectangular tiling, dual-scale + shift parameterization has a small additional overhead compared to single-scale + shift parameterization; the total auxiliary parameters are $2 \times N \times M/T + M$.

### 2.2 REPRESENTATION SPACE

Before assigning values to the parameters from which we will reconstruct our matrix, we may want to transform the space in which the matrix is represented, to make the reconstruction better aligned with some quality metric (like weight MSE or end-to-end accuracy on some validation data). The two most common among such transformations of the weight space are rotations (like the Hadamard transform (Ashkboos et al. (2024)), or even learned rotations (Liu et al.)), and channel-wise scaling (like in activation aware quantization (AWQ, Lin et al. (2024b)) or Smoothquant (Xiao et al. (2023))). Here, we propose a new transformation of the weight matrix using our dual-scaling parameterization.

### 2.2.1 PROXY METRIC AND SINKHORN NORMALIZATION

First, let us give an intuition of why dual-scaling is useful. Our dual-scaling representation offers a kind of flexibility in parameter assignment missing in other formats (e.g., Eq. 1): In 'single scaling'

---

**Algorithm 1** SINQ: Alternatingly normalize the standard deviation of the rows and columns of the matrix to be quantized. Then apply a standard quantization method (e.g., RTN).

---

**Require:** $\mathbf{W} \in \mathbb{R}^{m \times n}$, niter, bits
**Ensure:** $\mathbf{Q} \in \mathbb{Z}^{m \times n}, \vec{s} \in \mathbb{R}^m, \vec{t} \in \mathbb{R}^n$
1: $\sigma_{\min} \leftarrow \min(\mathbf{W}.\text{std}(\text{dim=0}).\min(), \mathbf{W}.\text{std}(\text{dim=1}).\min())$
2: $\hat{\mathbf{W}} \leftarrow \mathbf{W}$
3: **for** $i \leftarrow 1$ to niter **do**
4: $\quad \vec{\sigma}_0 \leftarrow \max(\hat{\mathbf{W}}.\text{std}(\text{dim=0}), \sigma_{\min})$
5: $\quad \hat{\mathbf{W}} \leftarrow \hat{\mathbf{W}}/\vec{\sigma}_0$
6: $\quad \vec{\sigma}_1 \leftarrow \max(\hat{\mathbf{W}}.\text{std}(\text{dim=1}), \sigma_{\min})$
7: $\quad \hat{\mathbf{W}} \leftarrow \hat{\mathbf{W}}/\vec{\sigma}_1$
8: **end for** $\qquad\qquad\qquad\qquad\qquad$ ▷ $\hat{\mathbf{W}}$ has std. dev. $\sigma_{\min}$ on all rows and columns
9: $\mathbf{Q}, \vec{z}, \vec{s} \leftarrow \text{Quantize}(\hat{\mathbf{W}}) \qquad\qquad\qquad$ ▷ omit $\vec{z}$ in case of symmetric quantization
10: **return** $\mathbf{Q}, \vec{z}, \vec{s} \odot \vec{\sigma}_1, \vec{\sigma}_0 \qquad$ ▷ the quantized matrix, optional shifts, and the two scale vectors

---

formats, an outlier at position $(i, j)$ necessarily causes all values either in column $i$ or row $j$ to have a higher error, because they share a large scale (that is needed to represent the outlier). With dual-scaling we may choose whether we distribute errors into column $i$ or row $j$ by assigning a higher scale either on the row or the column (see Fig. 1 for an illustration).

To find scale factors that balance the impact of outliers between rows and columns, we propose to minimize what we term the imbalance of the matrix. We define the imbalance $I$ as

$$I(\mathbf{W}) = \frac{\vec{\sigma}_{\max}(\mathbf{W})}{\vec{\sigma}_{\min}(\mathbf{W})} = \frac{\max_{i \in \{0,1\}} [\mathbf{W}.\text{std}(\text{dim=}i).\max()]}{\min_{i \in \{0,1\}} [\mathbf{W}.\text{std}(\text{dim=}i).\min()]}, \tag{4}$$

where $\vec{\sigma}_{\max}(\mathbf{W})$ is the maximum across the standard deviations of all rows and columns of the matrix and $\vec{\sigma}_{\min}(\mathbf{W})$ the corresponding minimum (in pseudo-pytorch notation).

Note that the matrix imbalance is inconvenient to optimize with gradient descent, because of the sparse gradients that result from the maximum and minimum operations. Instead, to find such doubly normalizing scale-factors, we propose a modified Sinkhorn-Knopp iteration (Sinkhorn & Knopp (1967)), where the goal is not to normalize all column and row sums (as in the standard algorithm), but all column and row standard deviations instead. The central idea is to alternatingly divide the rows and columns by their current standard deviations, see Alg. 1. Note that, in practice, we accumulate the scale factors in the log-domain for numerical stability, clip update values to avoid large jumps, and implement an early-stopping measure that keeps track of the imbalance. Further details are given in the supplementary code. We term this approach Sinkhorn Normalized Quantization (SINQ).

Akhondzadeh et al. suggest the kurtosis as a local proxy metric and optimization target for making matrices more easily quantizable, in the context of finding optimal rotations to apply to each layer. We find that 1) our imbalance optimization substantially reduces the average kurtosis of both rows and columns and 2) that directly minimizing kurtosis (while increasing imbalance) in our setting decreases end-to-end accuracy, see Fig. 2. This indicates that *for the dual-scaling setting*, imbalance is a better proxy target for ease of quantization than kurtosis.

### 2.2.2 ACTIVATION-AWARE CALIBRATION: FROM AWQ TO A-SINQ

AWQ (Lin et al. (2024b)) finds a vector of scales for each input of a linear layer, by minimizing the 2-norm between the linear layers output with the original and the scaled, quantized weight matrix. Formulaically,

$$\alpha^* = \arg\min_{\alpha} \left\| \vec{x} \cdot \mathbf{W}^T - \vec{x}/\vec{\mu_x}^{\alpha} \cdot \vec{x} \cdot d_q(q(\vec{\mu_x}^{\alpha} \odot \mathbf{W}))^T \right\|_2, \tag{5}$$

where $\vec{x}$ is a set of inputs, $\vec{\mu_x}$ is the sample mean of the absolute value of $\vec{x}$, $q(\cdot)$ is the quantization function, $d_q(\cdot)$ is the dequantization function and $\alpha^*$ is a per-layer parameter (a scalar).[1]

---

[1] For results in combination with our method, we modify this formula by changing the norm to a 1-norm, which we observe to give slightly better results in combination with SINQ.

Notably, AWQ scaling can be combined with SINQ. However, a naïve approach does not work. Suppose we feed an awq-scaled matrix into the SINQ algorithm. In that case, the iterated normalization can remove the awq-scales. Instead, we first normalize the matrix as in Alg. 1, then scale the normalized matrix with the awq-scales, and finally quantize. In this ordering of operations, the awq-scales fulfill their purpose of weighting matrix entries by importance. The awq-scales can be absorbed into one of the dual-scales.

## 2.3 Implementation Considerations

When using 1D tiling, the second scale $\vec{t}$ can be applied as a scale vector to the input of the quantized linear layer, rather than when reconstructing the weight (see Eq. 6). In this formulation, the forward complexity of the dual-scaling approach becomes very similar to AWQ: The term inside the square bracket is the RTN dequantization, and for each linear layer, we need to do one additional element-wise scaling of activations (just like in AWQ).

$$
\begin{aligned}
\vec{x} \cdot \mathbf{W}_{\text{approx}}^T &= \vec{x} \cdot \left[ \vec{s} \odot (\mathbf{Q} + \vec{z}) \odot \vec{t} \right]^T \\
&= \left( \vec{x} \odot \vec{t} \right) \cdot \left[ \vec{s} \odot (\mathbf{Q} + \vec{z}) \right]^T
\end{aligned}
\tag{6}
$$

The overhead of doing the additional scaling is small in practice, see Sec. 3.4 .

### 2.3.1 No-Overhead SINQ

To avoid the small overhead of the additional element-wise scaling, we can absorb these scales into preceding layers. This comes with the caveat that for many commonly used models, some layers need to share this second scale. In the experiments, we show that this implies a trade-off between output quality (Appendix A.3) and a minor inference time overhead (see Sec. 3.4).

## 3 Experiments

We evaluate our proposed methods against several strong baselines in 4-bit (and to a lesser extent 3-bit) quantization using the permissively licensed and powerful Qwen3 family of models by Yang et al. (2025). We use the evaluation settings of Zheng et al. (2025). In accordance with Dutta et al. (2024), we report perplexities for language modeling and flip percentages for QA tasks. Flip percentages indicate how often the quantized model predicts a different result from the original full-precision model. Additionally, benchmark results for reasoning benchmarks are provided in the appendix. Code to reproduce the perplexities reported for our methods in this section can be found in the supplementary.

We highlight here that our method and implementation are architecture agnostic; i.e., there is no interdependency between the quantization of different layers (unlike, e.g., in methods using Hadamard transformations). For all models we tried, it works out of the box.

Wherever there is no mention to the contrary, we set the group size to 64, batch-size to 8, and for SINQ use 1D tiling and dual-scaling + shift parameterization.

To account for the overhead of different parameterizations and tiling strategies fairly, in our experiments, we report the total memory use (including activations) and look for Pareto-optimal parameterizations in the output quality vs. memory trade-off.

### 3.1 Uncalibrated Uniform Quantization

In Tab. 1, our method outperforms the baselines in every uncalibrated case in terms of C4 (Raffel et al. (2020)) and WikiText2 perplexity, sometimes reducing the residual difference to the 16-bit baseline by more than half. Similarly, our method performs best in terms of the average number of flips (see Tab. 2). Fig. 3 shows the memory-perplexity Pareto plot for different quantization methods across a wide range of Qwen3 models. Because the Qwen3 models are available in many different sizes, our method can dominate the bfloat16 baselines across a large range of available memory, from ca. 1.5 GB to 65 GB. Some additional perplexity results, including on Llama models (Sec. A.6), DeepSeek-V3 (Sec. A.7), and Mixture-of-Experts (MoE, Fedus et al. (2022)) models (Sec. A.14).

Table 1: Weight-only uncalibrated uniform PTQ on Qwen3 models with 3-bit and 4-bit quantization, reporting perplexity and actual memory usage (GB). Lower is better for all metrics. The best result for a given setting is marked in **bold**.

| | Method | Qwen3-1.7B | | | Qwen3-14B | | | Qwen3-32B | | |
| | | Mem. | *Wiki2*↓ | *C4*↓ | Mem. | *Wiki2*↓ | *C4*↓ | Mem. | *Wiki2*↓ | *C4*↓ |
|---|---|---|---|---|---|---|---|---|---|---|
| | Original (BF16) | 3.44 | *16.67* | *19.21* | 29.54 | *8.64* | *12.01* | 65.52 | *7.60* | *10.77* |
| 3-BIT | RTN[†] | 1.28 | 32.43 | 31.10 | 9.23 | 10.50 | 14.88 | 17.61 | 30.78 | 35.83 |
| | Hadamard + RTN[†] | 1.28 | 32.40 | 31.07 | 9.23 | 10.60 | 15.10 | 17.61 | 11.26 | 14.83 |
| | HQQ | 1.28 | 32.10 | 30.54 | 9.23 | 10.73 | 14.39 | 17.62 | 9.09 | 12.58 |
| | **SINQ** *(ours)* | 1.28 | **22.39** | **24.88** | 9.25 | **9.33** | **12.90** | 17.61 | **8.79** | **11.83** |
| 4-BIT | RTN[†] | 1.42 | 18.74 | 20.81 | 10.54 | 8.95 | 12.50 | 20.78 | 8.92 | 12.80 |
| | Hadamard + RTN[†] | 1.42 | 19.10 | 20.70 | 10.54 | 8.85 | 12.35 | 20.78 | 8.28 | 11.60 |
| | HQQ | 1.42 | 18.96 | 22.10 | 10.54 | 8.78 | 12.36 | 20.78 | 8.62 | 12.20 |
| | **SINQ** *(ours)* | 1.42 | **17.14** | **19.83** | 10.56 | **8.76** | **12.21** | 20.73 | **7.74** | **10.96** |

[†] Baseline result obtained by running our own implementations.

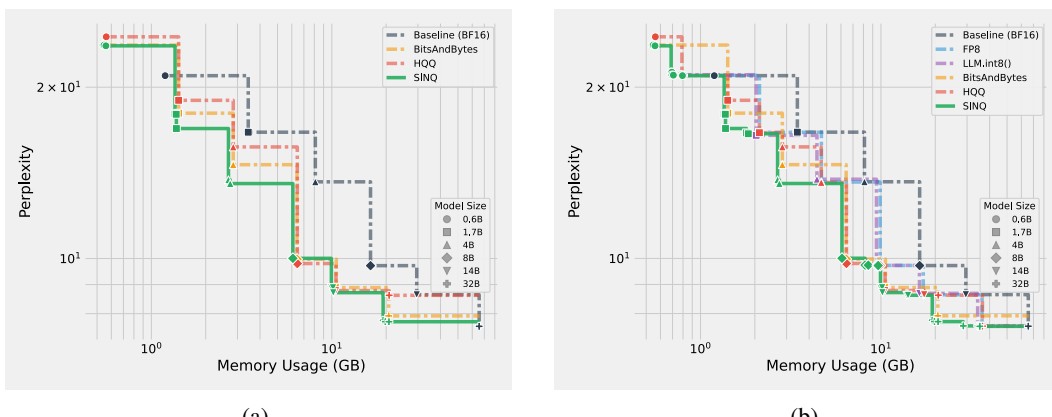

(a)                                                                (b)

Figure 3: Pareto plot in terms of memory vs. WikiText2 perplexity for Qwen3-0.6B to 32B for different uncalibrated quantization methods. (a) compares different 4-bit methods (including FP4, INT4, and NF4 where available). The maximum distance from the 4-bit pareto front of our method is $< 0.01$ppl. Note that the difference to the baseline is small. (b) allows bit widths of 4, 6, 8. For 8-bit quantization we inlcude `LLM.int8()` from Dettmers et al. (2022) as a reference method. Both plots include the BF16 model as a baseline. For these plots we allow group sizes 64 and 128 for all methods.

### 3.1.1 RESULTS ON LARGE MODELS

We further evaluate our method on two large models, Qwen3-235B-A22B by Yang et al. (2025) and DeepSeek-V2.5-236B DeepSeek-AI (2024), see Tab. 3. Notably, these are both MoE models, and the latter uses Multi-head Latent Attention (MLA). This underlines the robustness of SINQ to different architectures.

### 3.2 UNCALIBRATED NON-UNIFORM QUANTIZATION

SINQ is compatible with non-uniform quantization levels, for example, NF4 as defined by Dettmers et al. (2023). In Tab. 4 we compare to various non-uniform 4-bit quantization methods. We simply replace the quantization function in Alg.1 with the NF4 quantizer. Also here the SINQ method improves over the NF4 baseline. We note that for the 32B model, SINQ with INT4 slightly outperforms SINQ with NF4.

Table 2: Flip rates (%) (as proposed by Dutta et al. (2024)) on HellaSwag, PIQA, and MMLU for Qwen3 models with 3-bit and 4-bit quantization. Lower is better. The best result for a given setting is marked in **bold**.

| | | | Qwen3-14B | | | | Qwen3-32B | | | |
|---|---|---|---|---|---|---|---|---|---|---|
| | | Method | *HellaSwag* | *PIQA* | *MMLU* | Avg.↓ | *HellaSwag* | *PIQA* | *MMLU* | Avg.↓ |
| CALIBRATION-FREE | 3-BIT | RTN[†] | 8.44 | 8.60 | 10.97 | 9.34 | 22.84 | 17.08 | 10.61 | 16.84 |
| | | Hadamard + RTN[†] | 10.68 | 10.93 | 16.21 | 12.60 | 19.83 | 13.17 | 12.81 | 15.27 |
| | | HQQ | 7.99 | 7.94 | 14.28 | 10.07 | 7.23 | 9.30 | 10.98 | 9.17 |
| | | **SINQ** *(ours)* | **5.34** | **7.02** | **10.82** | **7.73** | **5.54** | **7.13** | **10.21** | **7.63** |
| | 4-BIT | RTN[†] | 2.92 | 4.57 | 4.89 | 4.13 | 4.18 | 6.31 | 5.28 | 5.26 |
| | | BnB (FP4) | 4.21 | 5.71 | 6.72 | 5.55 | 12.32 | 9.14 | 6.25 | 9.24 |
| | | BnB (NF4) | 2.66 | **3.10** | 4.70 | 3.49 | 3.73 | 3.48 | 4.76 | 3.99 |
| | | Hadamard + RTN[†] | 3.63 | 5.55 | 4.88 | 4.69 | 4.01 | 6.02 | 5.32 | 5.12 |
| | | HQQ | 2.81 | 4.35 | 5.17 | 4.11 | 5.83 | 5.18 | 4.98 | 5.33 |
| | | **SINQ** *(ours)* | **2.36** | 3.37 | **4.65** | **3.46** | **2.52** | **3.59** | **4.69** | **3.60** |
| CALIBRATED | 3-BIT | GPTQ | 5.18 | 7.83 | 11.17 | 8.06 | 6.33 | 8.76 | 10.25 | 8.45 |
| | | Hadamard[†] + GPTQ | 5.14 | 7.56 | 11.15 | 7.95 | 5.52 | 8.71 | **10.08** | 8.10 |
| | | **A-SINQ** *(ours)* | **5.13** | **7.18** | **10.36** | **7.56** | **5.23** | **7.62** | 10.15 | 7.67 |
| | 4-BIT | GPTQ | 2.24 | 4.13 | 4.56 | 3.64 | 2.78 | **3.48** | 4.80 | 3.69 |
| | | Hadamard[†] + GPTQ | 2.22 | 3.54 | 4.53 | 3.43 | 2.70 | 3.54 | 4.79 | 3.68 |
| | | AWQ | 2.23 | 3.26 | **4.10** | 3.20 | 2.59 | 4.13 | 4.44 | 3.72 |
| | | **A-SINQ** *(ours)* | **2.20** | **3.11** | 4.23 | **3.18** | **2.57** | 3.86 | **4.38** | **3.60** |

[†] Baseline result obtained by running our own implementations.

Table 3: Weight-only PTQ on **DeepSeek-V2.5-236B** and **Qwen3-235B-A22B** MoE models with 3-bit and 4-bit quantization, reporting perplexity and actual memory usage (GB). Lower is better for all metrics. The best result for a given setting is marked in **bold**.

| | | DeepSeek-V2.5-236B | | | Qwen3-235B-A22B | | |
|---|---|---|---|---|---|---|---|
| Setting | Method | Mem. | *Wiki2*↓ | *C4*↓ | Mem. | *Wiki2*↓ | *C4*↓ |
| Baseline | Original (BF16) | 471.56 | 5.36 | 8.15 | 470.19 | 5.37 | 9.30 |
| Calibration-free (3-bit) | RTN | 110.90 | 5.91 | 8.84 | 110.98 | 10.11 | 13.92 |
| | HQQ | 110.92 | 5.89 | 8.76 | 114.43 | 13.07 | 16.38 |
| | **SINQ** *(ours)* | 110.91 | **5.82** | **8.74** | 110.99 | **6.27** | **10.03** |
| Calibration-free (4-bit) | RTN | 134.24 | 5.49 | 8.27 | 134.03 | 5.65 | 9.49 |
| | BnB (FP4) | 134.52 | 5.55 | 8.41 | 134.10 | 6.67 | 10.21 |
| | BnB (NF4) | 134.52 | 5.49 | 8.28 | 134.10 | 5.60 | 9.49 |
| | HQQ | 134.25 | 5.49 | 8.27 | 134.03 | 5.60 | 9.46 |
| | **SINQ** *(ours)* | 134.51 | **5.48** | **8.25** | 134.06 | **5.58** | **9.43** |

## 3.3 CALIBRATED UNIFORM QUANTIZATION

To demonstrate compatibility with calibration approaches, in Tab. 5 we consider the combination of SINQ and AWQ (see Sec. 2.2.2 for the methodology). For a better match to the original AWQ implementation, we quantize our $\vec{s}, \vec{z}$ to 8 bits in these calibrated experiments. In several cases, even our uncalibrated method outperforms the calibrated baselines, but the addition of AWQ calibration brings further improvements.

## 3.4 INFERENCE TIME

The inference time of SINQ-quantized models in the default 1D-tiling case is very close to, or identical to, that of models quantized with standard 1D-tiled methods like HQQ or GPTQ. Specifically, the no-overhead formulation of SINQ (see Sec. 2.3.1) achieves identical inference time.

Table 4: Weight-only uncalibrated PTQ on Qwen3 models with 4-bit non-uniform quantization, reporting perplexity and actual memory usage (GB). Lower is better for all metrics. The best *non-uniform* result for a given setting is marked in **bold**, the results where SINQ with uniform quantization outperforms the non-uniform baselines are marked red.

| | Method | Qwen3-1.7B | | | Qwen3-14B | | | Qwen3-32B | | |
|---|---|---|---|---|---|---|---|---|---|---|
| | | Mem. | *Wiki2↓* | *C4↓* | Mem. | *Wiki2↓* | *C4↓* | Mem. | *Wiki2↓* | *C4↓* |
| | Original (BF16) | 3.44 | *16.67* | *19.21* | 29.54 | *8.64* | *12.01* | 65.52 | *7.60* | *10.77* |
| 4-BIT | BnB (FP4) | 1.42 | 24.05 | 23.44 | 10.59 | 8.88 | 12.54 | 20.67 | 11.93 | 16.90 |
| | BnB (NF4) | 1.42 | 18.00 | 20.43 | 10.59 | 8.89 | 12.27 | 20.67 | 7.94 | 11.21 |
| | HIGGS (non-uniform) | 1.51 | 23.98 | 25.27 | 10.28 | 9.13 | 12.56 | 19.88 | 8.02 | 11.24 |
| | **SINQ** (NF4) *(ours)* | 1.42 | **16.94** | **19.83** | 10.56 | **8.72** | **12.13** | 20.73 | **7.83** | **10.97** |
| | **SINQ** *(ours, uniform)* | 1.42 | 17.14 | 19.83 | 10.56 | 8.76 | 12.21 | 20.73 | 7.74 | 10.96 |

Table 5: Weight-only PTQ on Qwen3 models with 3-bit and 4-bit quantization, reporting perplexity and actual memory usage (GB). Lower is better for all metrics. The best result for a given setting is marked in **bold**, the *calibration-free* results that outperform all calibrated baselines at equal bits (other than our own) are marked red.

| | Method | Qwen3-1.7B | | | Qwen3-14B | | | Qwen3-32B | | |
|---|---|---|---|---|---|---|---|---|---|---|
| | | Mem. | *Wiki2↓* | *C4↓* | Mem. | *Wiki2↓* | *C4↓* | Mem. | *Wiki2↓* | *C4↓* |
| | Original (BF16) | 3.44 | *16.67* | *19.21* | 29.54 | *8.64* | *12.01* | 65.52 | *7.60* | *10.77* |
| 3-BIT | GPTQ | 1.26 | 32.21 | 31.05 | 9.28 | 9.54 | 13.03 | 17.70 | 9.03 | 12.38 |
| | Hadamard[†] + GPTQ | 1.26 | 24.70 | 25.37 | 9.28 | 9.61 | 12.92 | 17.70 | 8.51 | 11.63 |
| | **A-SINQ** *(ours)* | 1.26 | **22.30** | **24.00** | 8.90 | **9.31** | **12.71** | 16.68 | **8.45** | **11.54** |
| | **SINQ** *(ours, calibration-free)* | 1.28 | 22.39 | 24.88 | 9.25 | 9.33 | 12.90 | 17.61 | 8.79 | 11.83 |
| 4-BIT | GPTQ | 1.38 | 19.70 | 21.51 | 10.24 | 8.81 | 12.22 | 19.99 | 7.80 | 10.99 |
| | Hadamard[†] + GPTQ | 1.38 | 18.12 | 20.38 | 10.24 | 8.81 | 12.19 | 19.99 | 7.78 | 10.95 |
| | AWQ | 1.38 | 16.90 | 19.95 | 10.25 | 8.78 | 12.24 | 20.00 | 7.79 | 10.96 |
| | **A-SINQ** *(ours)* | 1.38 | **16.67** | **19.73** | 10.21 | **8.71** | **12.13** | 19.83 | 7.78 | **10.93** |
| | **SINQ** *(ours, calibration-free)* | 1.42 | 17.14 | 19.83 | 10.58 | 8.76 | 12.21 | 20.73 | 7.74 | 10.96 |

[†] Baseline result obtained by running our own implementations.

For the standard SINQ formulation, we compare the inference time of a HQQ-quantized linear layer using the **gemlite** kernel Badri et al. (2024) with that of a SINQ-quantized layer. For the latter, we naively implement the second scale using a PyTorch element-wise multiply before applying the kernel. As shown in Tab. 6, this incurs less than $2\%$ overhead.

## 3.5 QUANTIZATION TIME

Quantization with SINQ is fast. On identical hardware, SINQ has an average runtime $1.1\times$ our RTN baseline. This is faster than the already efficient HQQ, at $> 2\times$, or calibrated methods like AWQ, at $> 30\times$ the RTN baseline. Further details are given in Tab. 10 and Fig. 5 in the appendix.

## 3.6 ABLATION STUDIES

In this section we compare several variants of our method, namely we compare the conditions 1) with and without shifts, 2) 1D and 2D tiling, 3) quantized (int8) and half precision (fp16) auxiliary variables. In Fig. 4, we see that in general, both tilings and precisions work well; differences are minor, and both settings have their sections of the Pareto front. The use of shifts does improve the Pareto front appreciably in some places. Based on these results, we choose a 1D tiling with shifts as a good default setting and quantize the auxiliaries to match the methods we are comparing against.

Table 6: Computational overhead of the additional scale in a naive implementation. We compare the matmul speed of the fast gemlite kernel for W4A16 operation with and without the additional scale as used by SINQ. In practice, this scale can often be absorbed into other operations to reduce overhead further.

| Batch Size | Input Dim | gemlite($\vec{x}$) [ms] | gemlite($\vec{x} \cdot \vec{t}$) [ms] | Naive Overhead [%] |
|:---:|:---:|:---:|:---:|:---:|
| 1 | 1024 | 0.0446 | 0.0454 | 1.8% |
| 1 | 2048 | 0.0448 | 0.0455 | 1.5% |
| 64 | 1024 | 0.0472 | 0.0476 | 0.8% |
| 64 | 2048 | 0.0479 | 0.0483 | 0.9% |

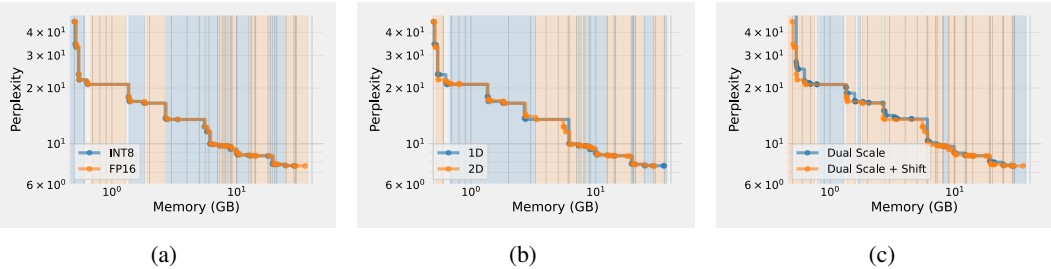

(a)            (b)            (c)

Figure 4: Ablation experiments in the form of memory-perplexity Pareto-fronts across the Qwen3 family. (a) Auxiliary variable precision (b) Tiling dimension (c) Using or not using shifts.

## 4 RELATED WORK

### 4.1 UNCALIBRATED, UNIFORM INTEGER QUANTIZATION

Most closely related to our approach are works focusing on quantization to uniform integer values without the use of a calibration set. Beyond the trivial (but effective) round-to-nearest (RTN) method with scales and shifts chosen to cover the full range of the input weights, there have been two major innovations in this domain. Firstly, half-quadratic quantization (HQQ, Badri & Shaji (2023)) proposes optimizing the values of the shifts found by RTN, so that a $p$-norm (usually $p = 0.7$) error between the original and the quantized matrix becomes minimal. Secondly, applying a Hadamard transform to all weights in a network has been observed to normalize the weight distributions (Tseng et al. (2024a)), which often eases quantization. The Hadamard approach has a high-level similarity to our approach, in that we also transform the weight matrices to find an easier-to-quantize format.

### 4.2 NON-UNIFORM QUANTIZATION

After training, neural network weights are usually not uniformly distributed. Therefore, quantization incurs lower errors when the quantization levels are also non-uniform, to match the distribution of the trained weights. Dettmers et al. (2023) proposes quantiles of the normal distribution as a preferable set of quantization levels resulting in the normal-float-4 (NF4) format (in the 4-bit case). The variance between optimal levels across different layers in a network is reduced when the weights of the network have been Hadamard transformed. This is used in HIGGS by Malinovskii et al. (2025) together with non-uniform quantization: Non-uniform quantization levels can be synergistic with weight matrix transformations. SINQ is orthogonal to the uniformity of the quantization levels; we show that it is compatible with non-uniform quantization in NF4-based experiments.

### 4.3 CALIBRATION

If quantization time and potential overfitting can be tolerated, using some data to calibrate the quantized value assignments can be a practical approach. A highly influential work is GPTQ Frantar et al. (2022) that considers the Hessian for a given layer to find weight pairs that can compensate for each other, if their quantization errors have opposite signs. A second approach, as seen in AWQ Lin et al. (2024b), is to minimize the prediction error of each linear layer (separately) under quantization (for

more details see Sec. 2.2.2). This per-layer prediction error minimization has been further developed by Shao et al. and Ma et al.. Similar to AWQ, CrossQuant Liu et al. (2024b) finds an input axis scale for the weight matrix with a calibration process. Elhoushi & Johnson (2025) combine non-uniform quantization with calibration to learn optimal non-uniform quantization levels. SINQ is orthogonal to calibration; we demonstrate its compatibility with calibration in AWQ-based experiments.

### 4.4 Weight Space Transformations

The concept of weight space transformation, such as applying the Hadamard transform, a random rotation, or scaling with a diagonal matrix, can be further improved by combining it with calibration and/or non-uniform quantization. HIGGS (Malinovskii et al. (2025)) applies Hadamard transforms and matches non-uniform quantization levels to the typically resulting distribution. QuaRot (Ashkboos et al. (2024)), SpinQuant (Liu et al.), and FlatQuant (Sun et al.) combine various calibration methods with rotations (including the Hadamard transform). Duquant (Lin et al. (2024a)) combines learned rotations with permutations for further flexibility. In Kurtail, Akhondzadeh et al. optimize rotations on a kurtosis proxy target. Several of these methods specifically target joint activation and weight quantization. The key differences to our method are that we use the dual-scaling and minimize the matrix imbalance, allowing the method to be uniform, calibration-free and, compared to rotated models, architecture agnostic (similar to HQQ (Badri & Shaji (2023)) and BnB (Dettmers et al. (2023))) in the sense that each linear layer can be treated independently (which is helpful for generalization to new architectures).

## 5 Conclusion

We have proposed using scaling factors in both matrix dimensions when representing weight matrices at low precision, along with an effective method for finding good values for these scaling factors, by simultaneously normalizing the row and column standard deviation through a modified Sinkhorn iteration. We show in numerous experiments that this method is fast and outperforms state-of-the-art methods for uniform quantization without calibration, and can be combined with widely used calibrated and/or non-uniform methods.

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

## A APPENDIX

### A.1 REPRODUCIBILITY STATEMENT

The code used to derive our LLM quantization results is given in the supplementary. This includes a full implementation of our method. For our key results, the perplexity evaluations, we use open-source code by Zheng et al. (2025) to ensure reproducible detail settings (e.g., context length). Our code, as well as the external code we base ours on, is permissively licensed to facilitate follow-up research. For our experiments, we use permissively licensed open-weight models to promote reproducibility further.

### A.2 RESULTS ON REASONING

In Tab. 7 we show results on reasoning benchmarks (Ye et al. (2025)). Here, we include the length of reasoning traces to ensure that lengthened reasoning does not negate some of the upside of quantization. Note that these are preliminary pass@1 results. These preliminary findings seem to suggest that the proposed method sustains robust reasoning capabilities while avoiding an increase in reasoning trace length, which is crucial for preserving the efficiency gains achieved through quantization.

Table 7: Reasoning performance on Qwen3-14B with 4-bit weight-only PTQ.

| | Method | Qwen3-14B | | | | | |
| | | AIME 2024 | | AIME 2025 | | Avg. | |
| | | Tok. | Acc. (%)↑ | Tok. | Acc. (%)↑ | Δ Tok. | Acc. (%)↑ |
|---|---|---|---|---|---|---|---|
| | Original (FP16) | *11 464* | *76.70* | *12 636* | *63.30* | *0* | *70.00* |
| CALIBRATION-FREE 4-BIT | RTN | 10 973 | 66.70 | 12 642 | 50.00 | -242 | 58.35 |
| | BnB (FP4) | 11 500 | 60.00 | 12 455 | 53.30 | -72 | 56.65 |
| | BnB (NF4) | 12 132 | 70.00 | 12 899 | 56.70 | +930 | 63.35 |
| | Hadamard + RTN | 11 210 | 70.00 | 12 989 | 53.30 | +99 | 61.65 |
| | HQQ | 11 862 | 70.00 | 12 991 | 56.70 | +367 | 63.35 |
| | **SINQ** | 11 660 | **73.30** | 12 305 | **63.30** | -67 | **68.30** |

### A.3 NO-OVERHEAD VARIANT

In Tab. 8, we show that the overhead-free formulation of SINQ also produces better quality outputs than comparable prior methods.

### A.4 COMBINATION WITH ACTIVATION QUANTIZATION

We consider the 1D tiled case where the input dimension remains ungrouped. Let $\mathcal{K}(\mathbf{x}_4, \mathbf{W}_4, s_w, z_w, s_x, z_x)$ denote a standard kernel for single-scale 4-bit matrix multiplication,

Table 8: Weight-only uncalibrated uniform PTQ on Qwen3 models with 4-bit quantization, reporting perplexity and actual memory usage (GB). Lower is better for all metrics. The best result for a given setting is marked in **bold**.

| Method | Qwen3-1.7B | | | Qwen3-14B | | | Qwen3-32B | | |
|---|---|---|---|---|---|---|---|---|---|
| | Mem. | *Wiki2↓* | *C4↓* | Mem. | *Wiki2↓* | *C4↓* | Mem. | *Wiki2↓* | *C4↓* |
| Original (BF16) | 3.44 | *16.67* | *19.21* | 29.54 | *8.64* | *12.01* | 65.52 | *7.60* | *10.77* |
| Hadamard + RTN[†] | 1.42 | 19.10 | 20.70 | 10.54 | 8.85 | 12.35 | 20.78 | 8.28 | 11.60 |
| HQQ | 1.42 | 18.96 | 22.10 | 10.54 | 8.78 | 12.36 | 20.78 | 8.62 | 12.20 |
| **SINQ** *(ours)* | 1.42 | **17.14** | **19.83** | 10.56 | **8.76** | **12.21** | 20.73 | **7.74** | **10.96** |
| **SINQ no overhead** *(ours)* | 1.42 | 17.63 | 19.99 | 10.56 | 8.78 | 12.32 | 20.73 | 7.78 | 11.15 |

[†] Baseline result obtained by running our own implementations.

where $\mathbf{x}_4$ and $\mathbf{W}_4$ represent the 4-bit quantized activations and weights, respectively, with corresponding scales $s$ and zero-points $z$. Under this formulation, the SINQ linear layer is expressed as $\mathcal{K}((\mathbf{x} \odot \mathbf{t})_4, \mathbf{W}_4, s_w, z_w, s_x, z_x)$. The key point is that the secondary scaling factor $\mathbf{t}$ is applied to the high-precision input $\mathbf{x}$ *before quantization*. In this way we can preserve the efficiency of standard 4-bit integer arithmetic kernels.

In Tab. 9 we see that SINQ still obtains a consistent improvement over RTN in this setting. More advanced methods, e.g., with SmoothQuant Xiao et al. (2023), will likely bring further gains in future work. Additional results in a W4A8 setting (without rotations) can be found in Sec. A.10.

Table 9: Wikitext perplexity comparison on Qwen-3 models using W4A4 quantization combined with an online block Hadamard rotation (block size 128) on activations. Lower is better. In bold is the best result.

| Method | Qwen-3 1.7B | Qwen-3 14B | Qwen-3 32B |
|---|---|---|---|
| | *Wiki2↓* | *Wiki2↓* | *Wiki2↓* |
| RTN | 35.63 | 10.55 | 9.65 |
| SINQ | **30.76** | **10.44** | **9.53** |

## A.5 TIMING RESULTS

In Tab. 10 we report quantization time results on a single GPU for various models. Although precise timings may vary with hardware, our method achieves times comparable to the RTN baseline and even surpasses HQQ, which is already regarded as a fast quantization technique. Furthermore, the calibrated version, A-SINQ, is substantially faster than popular state-of-the-art calibrated methods like GPTQ and AWQ. Fig. 5 shows the distribution of quantization times over 10 runs for various popular quantization methods on Qwen3-32B on GPU.

Table 10: Average quantization time (seconds) across 10 runs for some Qwen3 models on GPU, comparing different quantization methods. The rightmost column reports the relative average slowdown with respect to RTN.

| Method | Qwen3-1.7B | Qwen3-4B | Qwen3-8B | Qwen3-14B | Qwen3-32B | Avg. cost |
|---|---|---|---|---|---|---|
| *RTN* | *2.91 s ±0.11* | *6.32 s ±0.06* | *11.35 s ±0.31* | *20.61 s ±0.87* | *46.79 s ±2.52* | *1.00×* |
| HQQ | 3.65 s ±0.13 | 10.15 s ±0.27 | 24.06 s ±1.54 | 43.62 s ±0.54 | 122.45 s ±2.45 | 2.32× |
| GPTQ | 193.33 s ±1.68 | 426.89 s ±0.75 | 669.06 s ±0.84 | 1160.37 s ±1.68 | 3064.62 s ±24.33 | 62.68× |
| AWQ | 104.63 s ±9.26 | 225.27 s ±3.91 | 392.51 s ±2.86 | 695.29 s ±1.19 | 1613.75 s ±9.79 | 34.46× |
| A-SINQ *(ours)* | 23.86 s ±0.17 | 49.81 s ±0.13 | 92.17 s ±0.33 | 173.93 s ±0.38 | 411.95 s ±0.57 | 8.54× |
| **SINQ** *(ours)* | **3.03 s ±0.29** | **6.33 s ±0.52** | **13.23 s ±0.64** | **21.38 s ±2.15** | **51.56 s ±2.00** | **1.09×** |

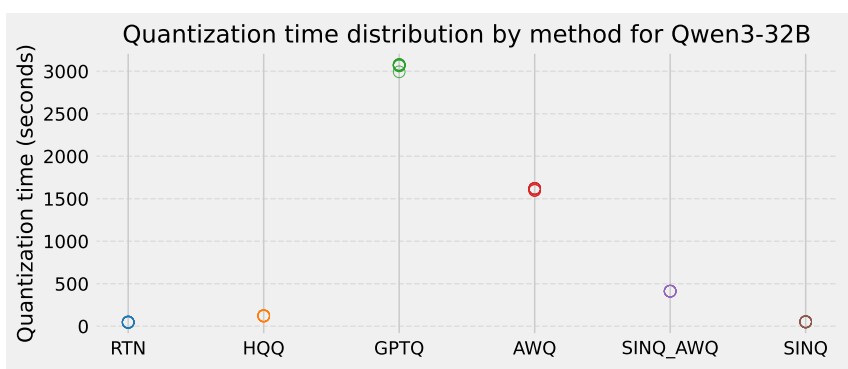

Figure 5: Distribution of quantization times for each method for Qwen3-32B.

## A.6 RESULTS ON LLAMA MODELS

In Tab. 11 we report quantization results on Llama family models. These findings further validate the effectiveness of SINQ also on this type of architecture.

Table 11: Weight-only PTQ on Llama models with 3-bit and 4-bit quantization, reporting perplexity and actual memory usage (GB). Lower is better for all metrics. In bold is the best result for a given setting.

| | Method | Llama 2-7B | | | Llama 3-8B | | | Llama 3-70B | | |
|---|---|---|---|---|---|---|---|---|---|---|
| | | Mem. | *Wiki2*↓ | *C4*↓ | Mem. | *Wiki2*↓ | *C4*↓ | Mem. | *Wiki2*↓ | *C4*↓ |
| | Original (BF16) | 14.08 | *5.47* | *6.90* | 17.45 | *6.13* | *9.61* | 141.11 | *2.86* | *7.30* |
| 3-BIT | RTN | 3.54 | 6.40 | 8.05 | 5.25 | 10.18 | 15.27 | 35.93 | 5.26 | 10.80 |
| | Hadamard + RTN | 3.54 | 6.31 | 7.89 | 5.25 | 9.97 | 15.25 | 35.93 | 4.99 | 10.45 |
| | HQQ | 3.62 | 7.05 | 9.03 | 5.24 | 9.55 | 14.68 | 36.16 | 85.64 | 23.32 |
| | **SINQ** *(ours)* | 3.54 | **6.14** | **7.72** | 5.35 | **8.04** | **12.32** | 35.93 | **4.52** | **8.48** |
| 4-BIT | RTN | 4.17 | 5.67 | 7.14 | 6.06 | 6.61 | 10.25 | 42.71 | 3.56 | 10.58 |
| | BnB (FP4) | 4.17 | 5.76 | 7.24 | 6.06 | 6.93 | 10.75 | 42.71 | 3.58 | 8.23 |
| | Hadamard + RTN | 4.17 | 5.65 | 7.10 | 6.06 | 6.72 | 10.23 | 42.71 | 3.54 | 9.95 |
| | HQQ | 4.22 | 5.68 | 7.13 | 6.06 | 6.58 | 10.22 | 42.71 | 3.26 | 8.13 |
| | **SINQ** *(ours)* | 4.19 | **5.60** | **7.04** | 6.06 | **6.53** | **10.14** | 42.81 | **3.17** | **7.51** |
| | BnB (NF4) | 4.17 | 5.65 | 7.09 | 6.07 | 6.56 | 10.20 | 42.71 | 3.22 | 7.68 |
| | **SINQ** (NF4) *(ours)* | 4.18 | **5.58** | **7.03** | 6.07 | **6.51** | **10.09** | 42.81 | **3.16** | **7.50** |

(Note: the left margin shows "CALIBRATION-FREE" spanning the 3-BIT and 4-BIT rows.)

## A.7 RESULTS ON DEEPSEEK-V3

In Tab. 12 we compare HQQ to SINQ on WikiText2 perplexity for DeepSeek-V3 Liu et al. (2024a).

Table 12: Weight-only PTQ on **DeepSeek-V3-685B** with 4-bit quantization. We report perplexity on WikiText-2 (lower is better). Best per setting in **bold**.

| Setting | Method | *Wiki2*↓ |
|---|---|---|
| Calibration-free (4-bit) | HQQ | 5.38 |
| | SINQ | **5.31** |

## A.8 ACCURACY RESULTS

In Fig. 6 and Tab. 13 we report accuracy results on various QA tasks. Note that flips (as reported in the main paper) are the more reliable (and less easily manipulated) metric than accuracy for

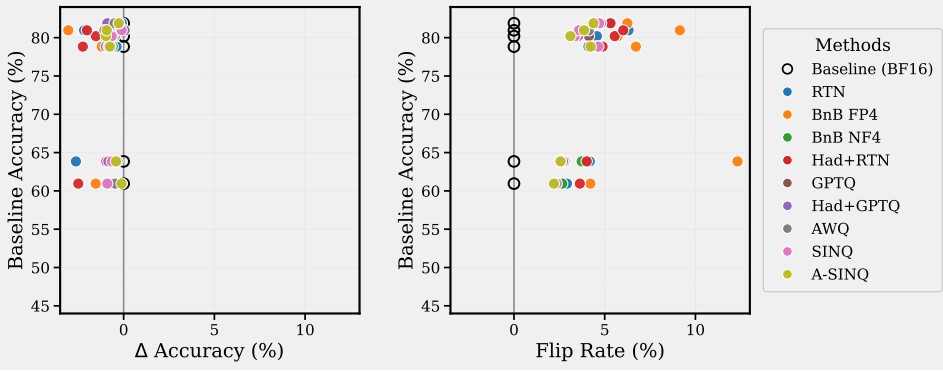

Figure 6: Comparison of baseline accuracy, accuracy changes, and flip rates across different 4-bit quantization methods (similar to Dutta et al. (2024)). On QA tasks, flips have been shown to be the more consistent quality metric of LLM quantization.

QA tasks, as shown in Dutta et al. (2024). Fig. 6 closely follows the analysis presented in prior work Dutta et al. (2024), further confirming the alignment of our findings with existing literature.

Table 13: Accuracy (%) on HellaSwag, PIQA, and MMLU for Qwen3 models with 3-bit and 4-bit quantization. Higher is better.

| | | Method | Qwen3-14B | | | | Qwen3-32B | | | |
| | | | HellaSwag | PIQA | MMLU | Avg.↑ | HellaSwag | PIQA | MMLU | Avg.↑ |
|---|---|---|---|---|---|---|---|---|---|---|
| | | Original (BF16) | 60.95 | 80.20 | 78.83 | 73.33 | 63.85 | 80.96 | 81.88 | 75.56 |
| CALIBRATION-FREE | 3-BIT | RTN | 56.99 | 77.80 | 75.01 | 69.93 | 46.98 | 71.82 | 78.53 | 65.78 |
| | | Hadamard + RTN | 49.66 | 73.45 | 67.53 | 63.55 | 50.43 | 75.41 | 78.10 | 67.98 |
| | | HQQ | 55.50 | **77.91** | 72.92 | 68.78 | 59.87 | 77.75 | 78.17 | 71.93 |
| | | **SINQ** *(ours)* | **58.03** | 77.20 | **75.82** | **70.35** | 60.65 | **79.49** | **78.89** | **73.11** |
| | 4-BIT | RTN | 60.11 | 79.11 | **78.44** | 72.55 | 61.22 | 78.78 | 81.78 | 73.93 |
| | | BnB (FP4) | 59.41 | 79.38 | 77.62 | 72.14 | 56.97 | 77.91 | 81.20 | 72.03 |
| | | BnB (NF4) | **60.47** | 79.71 | 78.23 | 72.80 | 63.12 | 79.98 | 81.60 | 74.90 |
| | | Hadamard + RTN | 58.45 | 78.67 | 76.58 | 71.23 | 62.90 | 78.94 | 80.99 | 74.28 |
| | | HQQ | 60.24 | 79.76 | 78.24 | 72.75 | 62.33 | 79.92 | **81.68** | 74.64 |
| | | **SINQ** *(ours)* | 60.05 | 79.54 | 78.00 | 72.53 | **63.20** | **80.85** | 81.63 | **75.23** |
| | | **SINQ** (NF4) *(ours)* | 60.35 | **79.72** | 78.37 | **72.81** | 63.18 | 80.52 | 81.32 | 75.00 |
| CALIBRATED | 3-BIT | GPTQ | **58.34** | 76.71 | 74.75 | 69.93 | 61.16 | 77.86 | 78.94 | 72.65 |
| | | Hadamard + GPTQ | 57.41 | **77.75** | 75.10 | 70.08 | 61.26 | 78.45 | 78.78 | 72.83 |
| | | **A-SINQ** *(ours)* | 58.16 | 77.15 | **75.40** | **70.24** | **61.47** | **79.22** | **79.00** | **73.23** |
| | 4-BIT | GPTQ | 60.55 | 79.43 | **78.11** | 72.70 | 63.22 | 80.20 | 81.36 | 74.93 |
| | | Hadamard + GPTQ | 60.27 | **79.60** | 77.85 | 72.57 | 63.01 | **81.01** | 80.98 | 75.00 |
| | | AWQ | 60.48 | 79.38 | 78.01 | 72.62 | **63.51** | 79.90 | 81.38 | 74.93 |
| | | **A-SINQ** *(ours)* | **60.84** | 79.22 | 78.07 | **72.71** | 63.43 | 80.03 | **81.61** | **75.02** |

## A.9 RESULTS ON PHI MODELS

In Tab. 14 we report quantization results on Phi family models. These findings further validate the effectiveness of SINQ also on this type of architecture.

Table 14: Weight-only PTQ on Phi models with 3-bit and 4-bit quantization, reporting perplexity and actual memory usage (GB). Lower is better for all metrics. In bold is the best result for a given setting.

| | | | **Phi-2** (3B) | | | **Phi-3** (4B) | | | **Phi-4** (15B) | | |
|---|---|---|---|---|---|---|---|---|---|---|---|
| | | Method | Mem. | *Wiki2*↓ | *C4*↓ | Mem. | *Wiki2*↓ | *C4*↓ | Mem. | *Wiki2*↓ | *C4*↓ |
| | | Original (BF16) | 5.18 | *9.82* | *13.83* | 7.11 | *6.01* | *8.96* | 27.31 | *6.67* | *11.13* |
| CALIBRATION-FREE | 3-BIT | RTN | 1.57 | 12.24 | 16.27 | 1.96 | 9.74 | 12.39 | 7.82 | 7.29 | 12.23 |
| | | HQQ | 1.57 | 11.37 | 15.69 | 1.96 | 11.60 | 16.42 | 7.82 | 7.41 | 15.60 |
| | | **SINQ** *(ours)* | 1.64 | **11.07** | **15.23** | 1.99 | 9.56 | 12.14 | 7.91 | **7.28** | **12.19** |
| | 4-BIT | RTN | 1.81 | 10.30 | 14.40 | 2.28 | 6.95 | 9.71 | 9.18 | 6.64 | 11.38 |
| | | HQQ | 1.81 | 10.09 | 14.23 | 2.28 | 6.85 | 9.70 | 9.18 | 6.80 | 14.84 |
| | | **SINQ** *(ours)* | 1.86 | **9.98** | **14.09** | 2.29 | 6.79 | 9.68 | 9.32 | **6.61** | **11.32** |

## A.10 COMPARISON TO CROSSQUANT

Here we compare to the CrossQuant method Liu et al. (2024b). We separate these results from the main text, because CrossQuant uses a W4A8G128 setting, so that the values are not directly comparable to the main results (using W4A16G64) of the paper. See Tab. 15

Table 15: Wikitext perplexity comparison to CrossQuant on Llama2 models (we use context length 2048, W4A8G128 to match reported CrossQuant results). Lower is better. In bold is the best result.

| | **Llama2-7B** | **Llama2-13B** |
|---|---|---|
| Method | *Wiki2*↓ | *Wiki2*↓ |
| Original (BF-16) | 5.47 | 4.88 |
| CrossQuant | 5.79 | 5.14 |
| ASINQ | **5.62** | **4.97** |

## A.11 COMPARISON TO CODE-BOOK-BASED METHODS

Here we compare to two recent code-book-based methods by Tseng et al. (2024b) and Tseng et al. (2024a). Note that code-book-based methods are incompatible with activation quantization and require non-standard operations / kernels (may not be NPU, TPU, mobile compatible). See Tab. 16.

Table 16: Wikitext perplexity comparison to code-book-based models on Llama2 models (context length 4096). Note that code-book-based methods are incompatible with activation quantization and require non-standard operations (may not be NPU, TPU, mobile compatible).

| | **Llama2-7B** | **Llama2-13B** |
|---|---|---|
| Method | *Wiki2*↓ | *Wiki2*↓ |
| Baseline | 5.12 | 4.57 |
| QTIP | **5.17** | **4.62** |
| QUIP# | 5.22 | 4.65 |
| ASINQ | 5.22 | 4.64 |

## A.12 FURTHER COMPARISON TO HIGGS

For a fairer comparison to the HIGGS method, in Tab. 17 compare it to SINQ with quantized auxiliaries (to ensure more similar memory usage).

Table 17: Comparison to HIGGS method with quantized auxiliary variables to better match the HIGGS memory use.

| | Method | Qwen3-1.7B | | | Qwen3-14B | | | Qwen3-32B | | |
|---|---|---|---|---|---|---|---|---|---|---|
| | | Mem. | *Wiki2↓* | *C4↓* | Mem. | *Wiki2↓* | *C4↓* | Mem. | *Wiki2↓* | *C4↓* |
| | Original (BF16) | 3.44 | *16.67* | *19.21* | 29.54 | *8.64* | *12.01* | 65.52 | *7.60* | *10.77* |
| 4-BIT | HIGGS (non-uniform) | 1.51 | 23.98 | 25.27 | 10.28 | 9.13 | 12.56 | 19.88 | 8.02 | 11.24 |
| | **SINQ** (NF4) *(ours)* | 1.42 | 16.94 | **19.83** | 10.56 | **8.72** | **12.13** | 20.73 | 7.83 | **10.97** |
| | **SINQ** (NF4) *(ours, q. aux.)* | **1.24** | **16.92** | 19.84 | **10.19** | **8.72** | **12.13** | **19.80** | **7.82** | 10.98 |

## A.13 COMBINATION OF SINQ AND HADAMARD ROTATION

We find that combining Hadamard and SINQ does not further improve results. Intuitively, this is because both Hadamard rotation and SINQ aim to transform the space in which we quantize the matrix – both succeed to some extent, with SINQ having an advantage, see Tab. 18.

Table 18: Performance comparison different space transformation methods (SINQ, hadamard) and their combination on Qwen3 models. Lower is better. In bold is the best result.

| | Qwen3-1.7B | Qwen3-14B | Qwen3-32B |
|---|---|---|---|
| Method | *Wiki2↓* | *Wiki2↓* | *Wiki2↓* |
| Hadamard+RTN | 19.10 | 8.85 | 8.28 |
| Hadamard+SINQ | 20.46 | 9.13 | 8.27 |
| SINQ | **17.14** | **8.76** | **7.74** |

## A.14 ADDITIONAL RESULTS ON MoE MODELS

In Tab. 19 we show some perplexity results on MoE models to underline the flexibility of our method. These results further demonstrate that SINQ is able to outperform state-of-the-art calibration-free methods for weight quantization.

Table 19: Weight-only PTQ on **DeepSeek-V2-Lite** and **Qwen3-30B-A3B** MoE models with 3-bit and 4-bit quantization, reporting perplexity and actual memory usage (GB). Lower is better for all metrics. In bold is the best result for a given setting.

| Setting | Method | DeepSeek-V2-Lite | | | Qwen3-30B-A3B | | |
|---|---|---|---|---|---|---|---|
| | | Mem. | *Wiki2↓* | *C4↓* | Mem. | *Wiki2↓* | *C4↓* |
| Baseline | Original (BF16) | 32.55 | *6.31* | *8.83* | 61.06 | *8.70* | *12.15* |
| Calibration-free (3-bit) | RTN | 9.12 | 7.94 | 10.98 | 15.10 | 12.28 | 15.89 |
| | HQQ | 9.12 | 8.36 | 11.74 | 15.10 | 10.52 | 14.39 |
| | **SINQ** *(ours)* | 9.02 | **7.45** | **10.32** | 15.13 | **10.19** | **13.62** |
| Calibration-free (4-bit) | RTN | 10.63 | 6.59 | 9.19 | 18.07 | 9.04 | 12.64 |
| | BnB | 10.63 | 6.82 | 9.49 | 18.08 | 9.68 | 12.93 |
| | HQQ | 10.85 | 6.61 | 9.18 | 18.07 | 9.14 | 12.64 |
| | **SINQ** *(ours)* | 10.50 | **6.49** | **9.07** | 18.13 | **9.02** | **12.41** |

