# OpenReview forum: "SINQ: Sinkhorn-Normalized Quantization for Calibration-Free Low-Precision LLM Weights"
_ICLR.cc/2026/Conference — Submitted to ICLR 2026_

### Official Review · Reviewer_jhSj · 2025-10-15

**Soundness:** 2
**Presentation:** 2
**Contribution:** 2
**Rating:** 2
**Confidence:** 4

**Summary:**

This paper presents SINQ, a post-training quantization method for large language models that uses dual-axis scaling and a modified Sinkhorn–Knopp algorithm to minimize "matrix imbalance," improving perplexity on models like Qwen3 and DeepSeek-V2.5 while being compatible with mainstream paradigms such as NF4 and AWQ. However, it suffers from critical flaws—including unclear experimental details, unproven core metrics, limited innovation, and missing key comparisons—resulting in a "Reject" score, though addressing these issues could lead to reconsideration.

**Strengths:**

1. The necessity of dual-axis scaling has not been verified: The paper fails to compare the performance of "row-only scaling", "column-only scaling", and "dual-axis scaling". This makes it impossible to prove the advantage of "dual-axis scaling" — for instance, if column-only scaling can achieve similar performance, the additional complexity of dual-axis scaling becomes meaningless.
2. It is a non-isolated solution and can be combined with mainstream quantization paradigms to expand application scenarios (e.g., NF4, AWQ).

**Weaknesses:**

1. Lack of experimental details: For the SINQ algorithm, the "specific selection rule for σmin" and "threshold setting for early-stopping" are not provided. In A-SINQ, the calibration dataset for AWQ (e.g., sample count, source) is not mentioned. While AWQ typically relies on 128–512 samples from the C4 dataset, the paper does not confirm consistency with this practice, nor does it explain whether calibration samples affect the optimization results of SINQ.

2. The rationality of using matrix imbalance as a surrogate metric is unproven: The paper defines matrix imbalance as $I(W)=\sigma_{min}(W)/\sigma_{max}(W)$ (the ratio of the minimum to maximum standard deviations of rows and columns) and claims that minimizing $I(W)$ improves quantization accuracy. However, it does not establish a mathematical connection between "matrix imbalance" and "quantization error" — for example, why a smaller $I(W)$ leads to lower post-quantization MSE or perplexity. The paper only observes through Figure 2 that "minimizing $I(W)$ reduces kurtosis", but fails to analyze the relationship between kurtosis and quantization error (e.g., whether reduced kurtosis necessarily decreases distribution overlap under low-bitwidth conditions), leaving the core assumption without theoretical support.

3. The paper modifies the standard Sinkhorn-Knopp algorithm to normalize row and column standard deviations, but does not prove the convergence of the modified algorithm (e.g., whether iterations enter cycles or if a unique fixed point exists). Additionally, it does not explain the basis for selecting the number of iterations $n_{iter}$ (e.g., why a fixed number is chosen instead of dynamic stopping based on the convergence threshold of $I(W)$), casting doubt on the algorithm’s stability. While Figures 2(a)(b) show that $I(W)$ stabilizes after 10 iterations, the paper does not explain "why 10 iterations are optimal" nor conduct ablation studies on the impact of $n_{iter}$ on performance.

4. The paper adopts the sequence "SINQ normalization → AWQ scaling → quantization" (Section 2.2.2). However, AWQ’s core lies in "activation-aware weight scaling", which relies on the distribution characteristics of original weights (e.g., correlation between activations and weights). Prior SINQ normalization alters the weight distribution, potentially disrupting the correlation relied on by AWQ. The paper does not compare the performance of alternative sequences such as "AWQ first, then SINQ" or "joint optimization of SINQ and AWQ", making it impossible to verify the rationality of the current sequence. It also fails to decompose contributions from "SINQ alone", "AWQ alone", and "their combination", leaving uncertainty about whether performance gains stem from dual-axis scaling or AWQ.

5. SINQ’s innovations are more akin to "combinatorial optimization of existing technologies" with limited breakthroughs: SINQ merely expands the scaling dimension from "weight-activation" or "single-axis weight" to "dual-axis weight", which is essentially an extension of the scaling target rather than a fundamental innovation. The standard Sinkhorn-Knopp algorithm normalizes row and column sums; SINQ only replaces the target with "row and column standard deviations" while retaining the algorithm framework (alternating iterative normalization), making this a routine modification rather than an innovative design.

6. Key comparative methods are missing: Representative methods such as FlatQuant and OSTQuant are not included in the comparisons.

7. The necessity of dual-axis scaling has not been verified: The paper fails to compare the performance of "row-only scaling", "column-only scaling", and "dual-axis scaling". This makes it impossible to prove the advantage of "dual-axis scaling" — for instance, if column-only scaling can achieve similar performance, the additional complexity of dual-axis scaling becomes meaningless.

8. For common datasets (HellaSwag, PIQA, MMLU), accuracy is the standard evaluation metric, while "Flip rates" are uncommon. The paper provides no justification for selecting this metric.

Based on the above weakness, I would assign a Reject score. I look forward to the authors addressing the aforementioned problems in future revisions, and I would be happy to reconsider and raise my score accordingly.

**Questions:**

See "Weaknesses"

---

> ### Author Response · Authors · 2025-11-20
>
> Thank you for your comments and suggestions. Indeed, comparison between dual- and single-scaling is extremely relevant. Therefore, let us first make an important clarification: **Almost all results in the paper are comparisons between setups using dual-scales (our proposed SINQ and A-SINQ) and setups using single scales (row/column scales)** (RTN, Hadamard+RTN, HQQ, AWQ, GPTQ, BnB NF4, BnB FP4, LLM.int8() and HIGGS). We will emphasize this in the paper in section 2.1.
>
> Here we address further comments:
> 1) **Lack of experimental details**: Some details are not described in the main paper text, however, **we supply the code** used to generate our results in the supplementary making our results fully reproducible. The mentioned **sigma_min is given in the paper in Alg. 1**. All details for our method can be found in the supplement including the calibration set (an Any4-style small set of diverse texts). For other methods we compare to, we use commonly used implementations (e.g. AutoAWQ for AWQ with the standard calibration set, a subset of C4), we will clarify this in the paper.
>
> 2) The **comparison between kurtosis and imbalance** minimization and their impact on end-to-end performance is **given in Figure 2**. We can see that optimizing kurtosis initially improves the quantization, but diverges soon, while the imbalance optimization remains stable.
>
> 3) Note that **Figure 2 c) shows the end-to-end performance as a function of number of sinkhorn iterations**. Concerning potential divergence, our implementation of the imbalance optimization does not diverge. We will further clarify the following in the paper: We always return the set of scales with minimal imbalance that was encountered before any step increases the imbalance.
>
> 4) **The comparison of SINQ vs. AWQ vs. SINQ + AWQ** (which we call A-SINQ) **is indeed already in the paper in table 2**. **We also explain why the opposite ordering of AWQ and Sinkhorn-Normalization does not make sense**: The normalization removes the AWQ scales in this case (see **first paragraph on page 5**).
>
> 5) The key question about adding a second scale, is how to use it well. This is the key novelty of our paper. We propose the **imbalance as a proxy target**, devise **a way to optimize it** and show in **numerous experiments** that this is useful in terms of output quality under weight quantization. Also note that **outperforming all previous comparable methods is in itself a contribution of great practical value**.
>
> 6) Here we provide some comparison **results for OSTQuant** on Llama models. We match or outperform these methods, while in contrast to these being 1) much faster at quantization time and 2) architecture agnostic (which allows trivial application to new architectures). FlatQuant is very heavy at quantization time using end-to-end optimization and also quantizes activations, which leads to much worse ppl.
>
>    To match OSTQuant paper experimental setting we use context length 2048 and W4A16.
>    | Model      | Method      | Wikitext     [ppl]  |
>    |------------|-------------|------|
>    | Llama2-7B  | Baseline    | 5.47 |
>    | Llama2-7B  | OSTQuant    | 5.64 |
>    | Llama2-7B  | ASINQ       | 5.58 |
>    ||||
>    | Llama2-13B | Baseline    | 4.88 |
>    | Llama2-13B | OSTQuant    | 4.94 |
>    | Llama2-13B | ASINQ       | 4.95 |
>    ||||
>    | Llama3-8B | Baseline    | 6.13 |
>    | Llama3-8B | OSTQuant    | 6.53 |
>    | Llama3-8B | ASINQ       | 6.53 |
>    ||||
>    | Llama3-70B | Baseline    | 2.86 |
>    | Llama3-70B | OSTQuant    | 3.19 |
>    | Llama3-70B | ASINQ       | 3.17 |
>    ||||
>    ||||
>
>
> 7) **Almost all results in the paper are comparisons between setups using dual-scales** (our proposed SINQ and A-SINQ) **and setups using single scales** (RTN, Hadamard+RTN, HQQ, AWQ, GPTQ, BnB NF4, BnB FP4, LLM.int8() and HIGGS). We will emphasize this in the paper.
>
> 8)  We will further clarify the use of Flips rather than accuracy in the paper. Note that a **recent NeurIPS paper ["Accuracy is not all you need"](https://arxiv.org/pdf/2407.09141) analysed this in detail and recommended Flips over accuracy** (we briefly **mention this in the caption of Table 2**, but will further explain). **Further, we do provide accuracy results in the appendix and a reproduction of key results of the mentioned paper** (confirming that on most popular datasets accuracies are not a meaningful metric). Our method outperforms the baselines also in terms of accuracy.
>
> **We hope this addresses your concerns. Please let us know if there are any open questions or further concerns left.**

---

> > ### Comment · Reviewer_jhSj · 2025-11-24
> >
> > Thank you for the authors' response. However, there remain **insurmountable gaps** in the key methodological foundations of this work, so I have decided to maintain my original score. The core points of my concerns are as follows:
> >
> > First, the **necessity of dual-axis scaling** lacks direct comparative verification; second, the **matrix imbalance as a proxy metric** is not supported by theoretical justification; third, the **modified Sinkhorn iteration** lacks a formal convergence proof; fourth, the **rationality of the combination with AWQ** has not been adequately analyzed; and finally, if scaling is applied to both rows and columns, **online scaling is required during actual inference**—only storage and bandwidth reduction can be confirmed, while computational acceleration remains uncertain.
> >
> > These all constitute fundamental issues regarding the method’s effectiveness and theoretical support. Unfortunately, the rebuttal only provides empirical explanations and supplementary experiments, without offering mathematical derivations, rigorous ablation studies, or theoretical basis to validate the correctness of the core mechanisms. Since these unresolved problems directly impact the method’s reliability, necessity, and innovativeness, the overall scheme still lacks sufficient academic rigor and persuasiveness. Therefore, I ultimately lean towards rejection.

---

> ### Author Response · Authors · 2025-11-27
>
> Let us discuss the perceived gaps one-by-one (Part 1/2):
>
> 1. >"First, the necessity of dual-axis scaling lacks direct comparative verification"
>
>      We compare the dual-axis scaling approach against **RTN, HQQ, AWQ, GPTQ, Hadamard+RTN, Hadamard+GPTQ, BnB (NF4), BnB (FP4), HIGGS, and OSTQuant**. All of **these techniques use single scaling**, or “row or column only scaling”, and we outperform them. Remember also that all these methods use tiling (see _2.1.1 Parameterization per tile_), with one scale e.g. every 64 rows/columns, whereas **our second scale is a single global scale that is independent of the tile size** (in all the relevant results, for which we use 1D tiling).
>
>      We include **more than 350 results** to compare single-scale with dual-scale methods using Qwen, DeepSeek, Llama, and Phi **models ranging from 1.7B to 685B** parameters, **across MoE and non-MoE architectures, models that use MLA and reasoning models**. In every case the results show that dual scaling improves the quantization quality. This goes beyond the standard evaluation breadth of many excellent papers on quantization.
>
>     **What more can we reasonably do to 'verify' that dual scaling helps improve the quantization quality?**
>
> 2. >"second, the matrix imbalance as a proxy metric is not supported by theoretical justification"
>
>     **In the context of proxy metrics for quantization, we are not aware of any case where a theoretical justification for improved *end-to-end performance* has been made**. Take e.g. the popular Hessian methods (GPTQ etc.) or Kurtosis based methods (Kurtail): These can offer good local intuitions about why they may be helpful, but a full end-to-end argument is precluded by the fact that the entire neural network is a highly non-linear system.
>
>     Such an intuition for the utility of SINQ is the following: In a linear layer, there is a scale ambiguity between the scales of the rows of its matrix and input scales. If we look only at weight-level MSE, we would consider small norm rows as unimportant, but these rows may well receive large inputs. Similarly, in an MSE perspective we would consider low norm columns unimportant, but they may get rescaled in a subsequent layer-norm layer. SINQ casts the weight matrix into a row / col - scale agnostic representation. This is useful to focus on the structure of the weight matrix independent of these misleading row and column scales (misleading in the aforementioned sense).

---

> ### Author Response · Authors · 2025-11-27
>
> Let us discuss the perceived gaps one-by-one (Part 2/2):
>
> 3. >"Third, the modified Sinkhorn iteration lacks a formal convergence proof"
>
>    The algorithm is implemented as a search procedure (see the code included in the submission and lines 194–197 in the paper). Since it is a search method, it cannot fail to converge, and the proof is straightforward. In the worst case, the search does not find any improvement over the original matrix in terms of imbalance. In that situation, we simply apply RTN to the original matrix, which produces results that are identical to standard RTN because the matrix remains unchanged.
>
> 4. > "fourth, the rationality of the combination with AWQ has not been adequately analyzed"
>
>     Subsection 2.2.2 discusses this (AWQ scales rows of the matrix by an importance weight derived from activations, to enhance their resolution (see original AWQ paper)). **The main point is that it is possible to define a calibrated version of SINQ**, which we call A-SINQ. There are many possible ways to construct such a calibrated variant, and in the paper we chose to present one that builds on AWQ. We include full results for this method and show that it outperforms AWQ, GPTQ and Hadamard+GPTQ in terms of quality, while also providing faster quantization. -- We are happy to give further explanations to any specific questions related to this (**i.e. what analysis are you missing?**).
>
> 5. > "and finally, if scaling is applied to both rows and columns, online scaling is required during actual inference—only storage and bandwidth reduction can be confirmed, while computational acceleration remains uncertain."
>
>     In response to Eu9q **we have produced measurements on this**.
>     1) using the Gemlite kernel, in a naive implementation (second scale added at pytorch level), the impact in terms of kernel speed is under 2% (c.f Eu9q for details).
>     2) for many architectures (incl. Qwen3, Phi-3 and Llama-3) **the second scale is not required at all during inference** since it can be entirely absorbed (c.f. Eu9q for details). We will explain this in the paper. In this zero-overhead formulation, SINQ is immediately compatible with frameworks such as SGLang and vLLM with the same inference speed as e.g. HQQ, but better end-to-end quality.
>
>     Additional note: **In many real deployments the system is memory-bound rather than compute-bound, so speed gains do not come from fast computation in kernels but from reduced data movement due to smaller, quantized weights**. This is especially true for small-batch local inference, where latency is dominated by loading weights, and for large MoE models, where even in the cloud very high batch sizes are needed before compute becomes the bottleneck. **In these bandwidth-limited regimes, reducing weight size translates directly into higher throughput.**

---

### Official Review · Reviewer_Eu9q · 2025-11-01

**Soundness:** 3
**Presentation:** 3
**Contribution:** 2
**Rating:** 4
**Confidence:** 4

**Summary:**

The paper proposes SINQ, a weight-only PTQ scheme that applies  dual scaling one scale per row and one per column, to each weight tile. The aim is to mitigate outliers along both dimensions and make ≤4-bit uniform quantization easier. The authors also introduce a proxy metric, matrix imbalance (the ratio of the largest to smallest row/column standard deviations), and a Sinkhorn–Knopp–style iteration that alternately normalizes row and column standard deviations to reduce this imbalance prior to quantization.

**Strengths:**

- **Simple, calibration-free** recipe that improves perplexity compared to strong uniform PTQ baselines at 3–4 bits across model sizes.
- Clear ablations comparing **imbalance** vs **kurtosis** as proxies for quantization difficulty.
- **Competitive results**, outperforming HQQ, GPTQ and AWQ in many settings.

**Weaknesses:**

- **Hardware evidence.** There are no end-to-end **inference throughput/latency** results or **kernel-level utilization** measurements; only **quantization-time** is reported. Without runtime data on common backends, deployment value is hard to assess.
- **Weight-only scope.** Although activation quantization is discussed, the experiments are weight-only. It remains unclear how dual scaling interacts with **W×A** low-precision matmuls and whether common kernel fusions remain intact.
- **Baselines.** Since the focus is weight quantization, a head-to-head with **codebook/rotation** approaches (e.g., **QuIP#**, **QTIP**) would strengthen the empirical case; these are mentioned in related work but not featured in the main tables.
- **Further empirical results.** Results would be more convincing with additional families (e.g., **LLaMA**, **Phi**) to demonstrate generality.
- CrossQuant appears closely related but is missing from the citations; it would help to explain the methodological differences and compare performance.

CrossQuant: A Post-Training Quantization Method with Smaller Quantization Kernel for Precise Large Language Model Compression., Liu, Wenyuan, et al. (2024).

**Questions:**

check the weakness,

1- how would dual scaling be implemented in hardware for W×A low-precision matrix multiplications, and what impact would it have inference speed?

---

> ### Author Response · Authors · 2025-11-20
>
> **Thank you** for your review and your helpful comments. Many of the points raised will help us improve the paper.
>
> 1) a) **Hardware evidence**: We evaluate a linear layer using the highly performant and widely used **gemlite kernel** (https://github.com/dropbox/gemlite/blob/master/README.md) comparing the standard single scale setting (as e.g. used by HQQ) and dual-scaling. We will add the code to the supplementary in the final version. **We observe an overhead of <2%** in this naive implementation. In some situations this overhead can be fully hidden by absorbing the second scale into a preceding layer on kernel level (e.g. into the SwiGLU layer).
>     | Batch size | Input dim | gemlite(x) [ms] | gemlite(s2*x) [ms] | Naive Overhead [%] |
>     |------------|-----------|------------|---------------|--------------|
>     | 1          | 1024      | 0.0446     | 0.0454        | 1.8%         |
>     | 1          | 2048      | 0.0448     | 0.0455        | 1.5%         |
>     | 64         | 1024      | 0.0472     | 0.0476        | 0.8%         |
>     | 64         | 2048      | 0.0479     | 0.0483        | 0.9%         |
>
>
> 1) b) We have implemented a **new, zero-overhead version of SINQ**. In this formulation we share input-side scales between parallel layers, so that these scales can then be absorbed into preceding-layer output-scales or normalization layer weights. This reduces the number of free auxiliary parameters leading to a slight drop in output quality, but has the big advantage that the resulting parameters can be used as a drop-in replacement for e.g. HQQ (--> immediate support with vLLM and similar frameworks). **The 'no overhead' still outperforms all baseline methods in terms of quality.**
>
>    | Model | Method             |  Wikitext ↓  Paper  [ppl]  |  Wikitext ↓   No overhead  [ppl] |
>    |-------|--------------------|------| ---- |
>    | Qwen-3 1.7B  | SINQ               | 17.14 | 17.63  |
>    | Qwen-3 1.7B  | ASINQ              | 16.67 | 16.67 |
>    |||||
>    |  Qwen-3  14B   | SINQ               | 8.76 | 8.78 |
>    | Qwen-3 14B   | ASINQ              | 8.71 | 8.76 |
>    |||||
>    | Qwen-3 32B   | SINQ               | 7.74 | 8.26 |
>    | Qwen-3 32B   | ASINQ              | 7.78 | 7.78 |
>    ||||
>    ||||
>
>    **In short, we can choose between a <2% overhead for best quality or a zero-overhead variant of SINQ for immediate integration in current frameworks (and a middle ground, where some of the overhead is absorbed without loss of quality). We will add these considerations to the paper.**
>
> 2) Thank you for the suggestion of combining SINQ with **activation quantization**. Let's consider the 1D tiled case, where the axis without grouping is the input. Assume we use a standard kernel for single-scale 4bit matmul K(x_4, W_4, sw, zw, sx, zx) (where x_4 is the 4bit activation, W_4 the 4bit weight and si and zi are the scales and zeropoints for x and W respectively). Then we can write the SINQ linear layer as K( (x * t)_4, W_4, sw, zw, sx, zx). The key point is: We need to 'absorb' the second scale t into the input x and then quantize the product x * t to 4 bits, instead of quantizing x directly to 4 bits, then we can do the matmul in 4 bits.
>
>    Here we report W4A4 results for SINQ and RTN combined with block hadamard rotation (block size 128) of the activations.
>
>     | Model | Bits  | Method |  Wikitext    [ppl]↓  |
>     |-------|-------|--------|-------|
>     | Qwen-3 1.7B  | W4A4  | RTN    | 35.63 |
>     | Qwen-3 1.7B  | W4A4  | SINQ   | 30.76 |
>     |||||
>     | Qwen-3 14B   | W4A4  | RTN    | 10.55 |
>     | Qwen-3 14B   | W4A4  | SINQ   | 10.44 |
>     |||||
>     | Qwen-3 32B   | W4A4  | RTN    |  9.65 |
>     | Qwen-3 32B   | W4A4  | SINQ   |  9.53 |
>     |||||
>     |||||
>
>     We still see a consistent improvement over RTN and **we don't see any problematic interaction of SINQ weight quantization and activation quantization**. We will add these results to the paper. **Note there are further W4A8 results in the comparison to CrossQuant**.

---

> ### Author Response · Authors · 2025-11-20
>
> 3) A key advantage over **codebook methods**, is that SINQ can be integrated with activation quantization as we see above. QUIP# and QTIP do not support this (and it is likely not possible to combine codebook based methods with low precision matmuls). A second advantage is that **SINQ uses readily available operations that various neural network accelerators support** (e.g. on mobile phones, TPUs, NPUs).
>
>    Given this, the methods are not comparable for many application scenarios. Nevertheless, for completeness, here are some weight-only comparisons to the strong results as published in the QTIP and QUIP# papers (in the no fine-tuning 4-bit setting). SINQ matches or outperforms the quality of QUIP#, and falls short in the second after comma digit of perplexities compared to QTIP:
>
>    To obtain results comparable to the QTIP paper, we use context length 4096 here.
>    | Model     | Method   |  Wikitext [ppl]  ↓   |
>    |-----------|----------|------|
>    | Llama2-7B | Baseline | 5.12 |
>    | Llama2-7B | QTIP     | 5.17 |
>    | Llama2-7B | QUIP#    | 5.22 |
>    | Llama2-7B | ASINQ    | 5.22 |
>    ||||
>    | Llama2-13B| Baseline | 4.57 |
>    | Llama2-13B| QTIP     | 4.62 |
>    | Llama2-13B| QUIP#    | 4.65 |
>    | Llama2-13B| ASINQ    | 4.64 |
>    ||||
>    ||||
>
>
>    We will add this table and discussion to the paper.
>
> 4) **Additional Models**: In the appendix, there are already some results on Llama models (Llama-2 7B, Llama-3 8B, Llama-3 70B). We will add further Llama (see other comments) and Phi results to the paper. Here are the Phi results. Note that AutoGPTQ and AutoAWQ do not currently support Phi models.
>
>     **Results for Phi-2 (3B):**
>     | Bit | Method | Memory | Wikitext     [ppl]↓ |
>     |-----|--------|--------|------------|
>     | BF16 | Original | 5.18 | 9.82 |
>     | _3-bit_ | RTN | 1.57 | 12.24 |
>     | _3-bit_ | HQQ | 1.57 | 11.37 |
>     | **_3-bit_** | **SINQ (ours)**| 1.64 | **11.07** |
>     |||||
>     | 4-bit | RTN | 1.81 | 10.30 |
>     | 4-bit | HQQ | 1.81 | 10.09 |
>     | **4-bit** | **SINQ (ours)** | 1.86 | **9.98** |
>     |||||
>     |||||
>
>     **Results for Phi-3 (4B):**
>     | Bit | Method | Memory | Wikitext    [ppl] ↓ |
>     |-----|--------|--------|------------|
>     | BF16 | Original | 7.11 | 6.01 |
>     | _3-bit_ | RTN | 1.96 | 9.74 |
>     | _3-bit_ | HQQ | 1.96 | 11.60 |
>     | **_3-bit_** | **SINQ (ours)** | 1.99 | **9.56** |
>     |||||
>     | 4-bit | RTN | 2.28 | 6.95 |
>     | 4-bit | HQQ | 2.28 | 6.85 |
>     | **4-bit** | **SINQ (ours)** | 2.29 | **6.79** |
>     |||||
>     |||||
>
>     **Results for : Phi-4 (15B)**
>     | Bit | Method | Memory | Wikitext     [ppl]↓ |
>     |-----|--------|--------|------------|
>     | BF16 | Original | 27.31 | 6.67 |
>     | _3-bit_ | RTN | 7.82 | 7.29 |
>     | _3-bit_ | HQQ | 7.82 | 7.41 |
>     | **_3-bit_** | **SINQ (ours)** | 7.91 | **7.28** |
>     |||||
>     | 4-bit | RTN | 9.18 | 6.64 |
>     | 4-bit | HQQ | 9.18 | 6.80 |
>     | **4-bit** | **SINQ (ours)** | 9.32 | **6.61** |
>     |||||
>     |||||
>
> 5) **CrossQuant**: Thank you for the reference, we will add this paper to the related work section. In brief: Similar to AWQ and our method, crossQuant also uses a second scale factor. In contrast to our approach, CrossQuant requires calibration of its parameters (like AWQ), CrossQuant does not make use of imbalance and does not normalize the standard deviation of the weights, and finally the accuracy comparison to calibrated SINQ (A-SINQ) is favourable for SINQ (here in the most comparable W4A8G128 setting)
>
>    To match CrossQuant paper experimental setting we use context length 2048 and W4A8G128.
>    | Model      | Method      |  Wikitext     [ppl]↓   |
>    |------------|-------------|------|
>    | Llama2-7B  | Baseline    | 5.47 |
>    | Llama2-7B  | CrossQuant  | 5.79 |
>    | Llama2-7B  | ASINQ       | 5.62 |
>    ||||
>    | Llama2-13B | Baseline    | 4.88 |
>    | Llama2-13B | CrossQuant  | 5.14 |
>    | Llama2-13B | ASINQ       | 4.97 |
>    ||||
>    ||||
>
>
> **Thank you again for your feedback. We hope that the many additional results and clarifications have addressed your concerns. Please don't hesitate with any further questions or if anything remains unclear or not satisfactorily addressed.**

---

### Official Review · Reviewer_uY9X · 2025-11-01

**Soundness:** 2
**Presentation:** 3
**Contribution:** 3
**Rating:** 6
**Confidence:** 5

**Summary:**

This paper presents a post-training weight quantization method on a dual-scale matrix quantization scheme.
Comparison to existing calibration-free PTQ methods including rotation-based methods is conducted empirically.

**Strengths:**

+ Clearly presented idea.
+ Potentially significant practical value.

**Weaknesses:**

- Practical overhead of dual-scaling on actual HW is not comprehensively discussed, except for memory efficiency.

**Questions:**

* I am not sure the comparison against Hadamard rotation, etc. is also based on dual-scale scheme or not--it should be for a fair comparison.
* Random rotation is purported to mix channels and thereby eliminate outliers--this seems to be doing similar things as dual-scale.  Could you do an ablation study with (1) Hadamard + single-scale, (2) Hadamard + dual-scale, and (3) SINQ?

---

> ### Author Response · Authors · 2025-11-20
>
> **Thank you** for taking the time to read our paper and for your review. The two points you raise are both very relevant and we are addressing them below. Please don't hesitate with any further questions!
>
> 1) **Overhead of dual scaling besides memory impact**. In short the compute overhead will be around **0% - 2%** (validated by a theoretical argument, a pessimistic measurement and a new variant of SINQ). We will add the following to the paper to better highlight the low impact of our method.
>
>     a) _Practical time measurement_: Our approach here is to take a fast kernel for 'single scale' dequantization and add the second scale factor on pytorch level. This will give us a simple, initial and somewhat pessimistic estimate of the performance impact (as we could do better by wrapping the second scale into the kernel). We use the **gemlite kernel** (link: https://github.com/dropbox/gemlite/blob/master/README.md) as a strong baseline. The results are in the following table. In short this pessimistic estimate yields a **<2% overhead**. Here is a detailed table:
>     | Batch size | Input dim | gemlite(x) [ms] | gemlite(s2*x) [ms] | Naive Overhead [%] |
>     |------------|-----------|------------|---------------|--------------|
>     | 1          | 1024      | 0.0446     | 0.0454        | 1.8%         |
>     | 1          | 2048      | 0.0448     | 0.0455        | 1.5%         |
>     | 64         | 1024      | 0.0472     | 0.0476        | 0.8%         |
>     | 64         | 2048      | 0.0479     | 0.0483        | 0.9%         |
>
>
> 	b) _Theoretical FLOPs analysis_: A theoretical approach is to simply look at the FLOPs in addition to the memory impact. This is useful to understand why the overhead we observe in measurement is so small. For a given matmul on x shaped 1x2048 and W shaped 2048x2048, we do 8.4 MFLOPs; for the added second scale we do 2.0 KFLOPs, i.e. in **raw FLOPs a <0.1% overhead**. All these operations scale equally with the batch size.
>
> 	c) _We've implemented a **new, zero-overhead version of SINQ**_. In this formulation we share input-side scales between parallel layers, so that these scales can then be absorbed into preceding-layer output-scales or layer-norm weights. This reduces the number of free auxiliary parameters leading to a slight drop in output quality, but has the big advantage that the resulting parameters can be used as a drop-in replacement for e.g. HQQ (--> immediate support with vLLM and similar frameworks). **The 'no overhead' still outperforms all baseline methods.**
>
>      | Model | Method             |  Wikitext ↓  Paper  [ppl]  |  Wikitext ↓   No overhead  [ppl] |
>    |-------|--------------------|------| ---- |
>    | Qwen-3 1.7B  | SINQ               | 17.14 | 17.63  |
>    | Qwen-3 1.7B  | ASINQ              | 16.67 | 16.67 |
>    |||||
>    | Qwen-3 14B   | SINQ               | 8.76 | 8.78 |
>    | Qwen-3 14B   | ASINQ              | 8.71 | 8.76 |
>    |||||
>    | Qwen-3 32B   | SINQ               | 7.74 | 8.26 |
>    | Qwen-3 32B   | ASINQ              | 7.78 | 7.78 |
>    ||||
>    ||||
>
>    **In short, we can choose between a <2% overhead for best quality or a zero-overhead variant of SINQ for immediate integration in current frameworks (and a middle ground, where some of the overhead is absorbed without loss of quality).**
>
> 2) **Combining Hadamard rotation and dual-scaling**. If we use dual-scaling, we need some way to choose the second scale -- the way to do this is SINQ, so we'll compare: Hadamard+RTN, SINQ, Hadamard+SINQ. What we find is that using Hadamard and SINQ together does not improve results further. Intuitively, this is because both Hadamard rotation and SINQ try to transform the space in which we do the quantization of the matrix -- both of them have some success in this, with SINQ having some advantage. We will add these results (on wikitext-2) and discussion to the paper.
>
>     | Method         | Qwen3-1.7B [ppl]↓  | Qwen3-14B [ppl]↓ | Qwen3-32B  [ppl]↓ |
>     |----------------|------------|-----------|-----------|
>     | Hadamard+RTN   | 19.10      | 8.85      | 8.28      |
>     | Hadamard+SINQ  | 20.46      | 9.13      | 8.27      |
>     | SINQ           | 17.14      | 8.76      | 7.74      |
>
>     In short, when SINQ is used, adding Hadamard rotations does not bring additional benefits for weight quantization.
>
> **Again, thanks a lot for your time and we're happy to address further questions and/or comments.**

---

### Author Response · Authors · 2025-12-03
**Rebuttal Summary for AC**

**Feedback from Reviewers ```uY9X``` and ```Eu9q```**

- Reviewers ```uY9X``` and ```Eu9q``` provided constructive feedback that helped
improve the paper. Their primary concern was the need to clarify the
**computational overhead** introduced by the second scale. In the revised
version, we now explicitly show that:

    1.  The theoretical overhead introduced by SINQ is below 0.1% FLOPs.
    2.  When implemented using a GemLite kernel and adding the scale at the
        PyTorch level, the measured overhead is under 2%, which is a
        pessimistic estimate since the scale can be fused into the kernel.
    3.  For many architectures (for example Qwen3-xB, Phi-3, Llama-2 and
        Llama-3), the second scale can be absorbed, yielding zero overhead
        in both compute and memory, with immediate compatibility with
        frameworks such as SGLang and vLLM.

    We also emphasize that in many real LLM deployments, systems are
    memory-bound rather than compute-bound. Speed improvements often stem
    not from faster kernels but from reduced data movement thanks to
    smaller, quantized weights. In bandwidth-limited regimes, reducing
    weight size directly increases throughput. This analysis is now included
    partly in Subsection ```2.3.1 No-overhead SINQ``` and Section  ```3.4 Inference time``` of the main paper and partly in ```Appendix A3```.

- Additional Comment from ```uY9X```: The reviewer asked about SINQ combined with Hadamard. We responded and added corresponding results in ```Appendix A.13```.

- Additional Comments from ```Eu9q```: the reviewer asked whether SINQ is compatible with activation
    quantization. Unlike codebook-based methods, which we had not
    covered, SINQ is compatible, and we now include supporting results
    for W4A4 and W4A8 in ```Appendix A.4``` and ```Appendix A.10```. These include comparisons with CrossQuant, a
    method that we were unfamiliar with but that SINQ outperforms (now in ```Appendix A.10```).
    We also added a comparison with codebook methods in ```Appendix A.11``` and
    updated the related works.

-   The reviewer  ```Eu9q``` requested if possible more experiments on Llama models, some of
    which were already present in the paper (Appendix A.6), and on Phi models. We added
    further results for Llama as well as Phi-2, Phi-3, and Phi-4 in
    ```Appendix A.9```, ```Appendix A.10```, ```Appendix A.11```.

**Feedback from Reviewer ```jhSj```**

**Most of the reviewer’s requests concerned results that were
already present in the submitted manuscript (in figures,
tables and algorithm environments)**. Their main claim, repeated in the rebuttal was that we did not compare row-only/column-only and dual-axis scaling. However, **this is the central part of our work** and nearly all baselines we
evaluate (RTN, Hadamard plus RTN, HQQ, AWQ, GPTQ, BnB NF4, BnB FP4,
LLM.int8, HIGGS) are precisely single-scale row or column methods.
We encourage the area chair to read our detailed exchange with the reviewer. Although we found the feedback uninformative, we nevertheless took the time to address every point carefully and thoroughly.

---

### Meta-Review · Area_Chair_oZVj · 2026-01-04

**Summary:**

The reviewers’ concerns that informed the suggested decision can be summarized as follows:

1. Reviewers questioned whether dual-axis scaling and Sinkhorn-style normalization represent a fundamentally new contribution, or a natural extension of existing scaling, rotation, and activation-aware quantization techniques (e.g., AWQ, SmoothQuant, Hadamard-based methods).

2. While memory savings and quantization-time costs are reported, reviewers were concerned about the absence of end-to-end inference throughput and latency results on standard backends, making it difficult to assess real-world deployment benefits.

3. Several reviewers requested clearer positioning relative to codebook- and rotation-based methods (e.g., QTIP, QUIP#, CrossQuant), which were either missing or only partially addressed in the original submission.

4. The method is primarily evaluated in a weight-only setting. Reviewers were unsure how dual scaling interacts with low-precision activation quantization and whether common kernel fusions remain valid.

**Reviewer Concerns:**

Concerns addressed by the rebuttal:

1. The authors provided both theoretical FLOP analysis and empirical kernel-level measurements, demonstrating <2% overhead in a pessimistic implementation and a zero-overhead variant through scale absorption.

2. New W4A4 and W4A8 experiments were added, showing consistent improvements over RTN and compatibility with activation quantization pipelines.

3. Results were added for LLaMA-2/3, Phi-2/3/4, and very large MoE models (Qwen3-235B, DeepSeek-V2.5), strengthening claims of architectural generality.

4. New comparisons and discussions involving CrossQuant, QTIP, and QUIP# were added, improving completeness.

Concerns still outstanding:

1. Despite the added experiments, the rebuttal does not fully dispel the impression that SINQ is an incremental refinement over existing scaling and normalization strategies rather than a fundamentally new principle.

2. Kernel microbenchmarks and theoretical arguments do not fully substitute for end-to-end inference throughput and latency evaluations on common LLM serving stacks, which remain absent.

3. While SINQ consistently improves over strong baselines, the gains are often modest, raising questions about whether the added methodological complexity justifies adoption in practice.

**Reviewer Scores:**

Reviewer uY9X (Initial rating: 6 Marginally above acceptance threshold)
Likely remain at 6. The rebuttal convincingly addresses overhead and comparison concerns, but novelty reservations would likely persist.

Reviewer Eu9q (Initial rating: 4 Marginally below acceptance threshold)
Likely unchanged, acknowledging added hardware analysis, activation-quantization results, and additional models, but still below acceptance due to deployment and novelty concerns.

Reviewer jhSj  (Initial rating: 2 reject)
Likely increase to 4.  Although many concerns are addressed, the core concern on the novelty and theoretical grounding seem not fully justified.

---

### Decision · Program_Chairs · 2026-01-26

Reject